# Older adults' experiences and perceived impacts of the Aging, Community and Health Research Unit-Community Partnership Program (ACHRU-CPP) for diabetes self-management in Canada: a qualitative descriptive study

Marie-Lee Yous [ID],[1] Rebecca Ganann [ID],[2] Jenny Ploeg [ID],[2] Maureen Markle-Reid,[2] Melissa Northwood [ID],[2] Kathryn Fisher,[2] Ruta Valaitis [ID],[2] Tracey Chambers [ID],[2] William Montelpare,[3] France Légaré [ID],[4,5] Ron Beleno,[2] Gary Gaudet,[3] Luisa Giacometti,[2] Deborah Levely,[2] Craig Lindsay,[2] Allan Morrison,[3] Frank Tang,[2] On behalf of the ACHRU-CPP Research Team[1]

For numbered affiliations see end of article.

**Correspondence to**
Dr Marie-Lee Yous;
yousm@mcmaster.ca

## ABSTRACT

**Objectives** To assess the experiences and perceived impacts of the Aging, Community and Health Research Unit-Community Partnership Program (ACHRU-CPP) from the perspectives of older adults with diabetes and other chronic conditions. The ACHRU-CPP is a complex 6-month self-management evidence-based intervention for community-living older adults aged 65 years or older with type 1 or type 2 diabetes and at least one other chronic condition. It includes home and phone visits, care coordination, system navigation support, caregiver support and group wellness sessions delivered by a nurse, dietitian or nutritionist, and community programme coordinator.

**Design** Qualitative descriptive design embedded within a randomised controlled trial was used.

**Setting** Six trial sites offering primary care services from three Canadian provinces (ie, Ontario, Quebec and Prince Edward Island) were included.

**Participants** The sample was 45 community-living older adults aged 65 years or older with diabetes and at least one other chronic condition.

**Methods** Participants completed semistructured postintervention interviews by phone in English or French. The analytical process followed Braun and Clarke's experiential thematic analysis framework. Patient partners informed study design and interpretation.

**Results** The mean age of older adults was 71.7 years, and the mean length of time living with diabetes was 18.8 years. Older adults reported positive experiences with the ACHRU-CPP that supported diabetes self-management, such as improved knowledge in managing diabetes and other chronic conditions, enhanced physical activity and function, improved eating habits, and opportunities for socialisation. They reported being connected to community resources by the intervention team to address social determinants of health and support self-management.

**Conclusions** Older adults perceived that a 6-month person-centred intervention collaboratively delivered by a team of health and social care providers helped support chronic disease self-management. There is a need for providers to help older adults connect with available health and social services in the community.

**Trial registration number** ClinicalTrials.gov ID: NCT03664583; Results.

## STRENGTHS AND LIMITATIONS OF THIS STUDY

⇒ This study included a rigorous qualitative design with a large sample size.
⇒ A rigorous analytical method was used involving multiple researchers with expertise in primary care, qualitative, ageing and diabetes research, as well as programme evaluation.
⇒ Patient and public research partners were involved in designing the intervention, informing the study design and interview guides, interpreting the results, and developing the manuscript.
⇒ A limitation of the study was related to the sample as there was a lack of cultural diversity with regard to ethnicity and under-representation of older adults from marginalised communities.

## INTRODUCTION

As of 2022, approximately 422 million people have been diagnosed with diabetes mellitus worldwide.[1] Older adults are more likely to have type 2 diabetes than younger adults[2] and are at risk for hypoglycaemia, which can adversely affect cognition, vision, hearing, mobility and mental health[3] as well as self-care activities including exercise and diet. More than 40% of older adults with diabetes have three or more chronic conditions,[4] including

hyperlipidaemia, hypertension, asthma, chronic obstructive pulmonary disease, chronic kidney disease, arthritis and heart failure.[5] Following management plans for one condition may be challenging due to symptoms or conflicting guidelines from another condition. Higher burden associated with the presence of multiple chronic conditions (MCC) has been linked to higher risk for mortality, decreased physical and mental functioning, and increased health services use.[2 6 7] Community-dwelling older adults with MCC are highly reliant on family/friend caregivers for support,[8] which can lead to poor mental and physical health, and financial losses among caregivers.[9] Caregivers' unmet needs[10] can lead to their increased use of hospital and emergency services.

Complex health interventions are defined as having multiple interacting components.[11] For older adults with diabetes, complex health interventions, such as peer support programmes, have demonstrated positive effects in managing their complex needs, sustaining lifestyle changes and achieving health benefits.[12–17] Complex interventions that target self-care and incorporate opportunities for peer-to-peer discussions among community-dwelling older adults can improve their mental and physical health and reduce falls.[18 19] Nurse-led self-management programmes for diabetes and other chronic conditions can lead to improvements in self-rated health, glycated haemoglobin (HbA1c) values, blood pressure, weight and self-management behaviours.[19 20] Providing diabetes self-management programmes through partnerships between primary care and community organisations (eg, senior centres, YMCA) supports programme uptake, implementation and sustainability,[12] which can lead to improved health literacy (ie, being able to locate, read and understand health information for informed decision-making). In a systematic review of randomised controlled trials (RCTs), health literacy was instrumental in enhancing diabetes knowledge, self-efficacy and physical activity.[21]

Receiving care from multiple providers from different health and community settings can lead to fragmented care for older adults, as referrals across organisations are often not well integrated.[22] Seamless care coordination and system navigation for older adults with MCC remain high priorities for this population. There is a need for innovative programmes for older adults that focus on the Quintuple Aims of high-quality care: (A) enhancing the patient experience of care, (B) creating healthy populations, (C) reducing healthcare costs, (D) improving the care delivery experience and (E) health equity.[23] Gaps in previous intervention designs include lack of emphasis on patient experience of care and considerations for health equity.[23]

The Aging, Community and Health Research Unit-Community Partnership Program (ACHRU-CPP) is a 6-month self-management intervention for community-living older adults (≥65 years old), diagnosed with type 1 or type 2 diabetes and at least one other chronic condition, and their family/friend caregivers (≥18 years old). The intervention was evaluated in a feasibility study in Ontario, Canada,[24] followed by a clinical trial in selected primary care and community settings in two Canadian provinces (Ontario

and Alberta).[25–27] A multijurisdictional pragmatic RCT is currently in progress to evaluate the implementation and effectiveness of the ACHRU-CPP in three Canadian provinces. To better understand how to address the complex needs of older adults, we sought to assess the experiences and perceived impacts of the ACHRU-CPP from the perspectives of older adults with diabetes and MCC.

## METHODS
### Design
This qualitative study is embedded within the multisite implementation-effectiveness type II hybrid RCT, as outlined in the protocol paper.[28] This study used a qualitative descriptive design, as described by Sandelowski,[29 30] to provide a fulsome summary while remaining close to the words of participants when describing their experiences with the ACHRU-CPP and its perceived impacts.

### ACHRU-CPP Program
The ACHRU-CPP is delivered by an interprofessional team of primary care providers, which includes a registered nurse (RN) and registered dietitian (RD) or nutritionist from a primary care setting, and a programme coordinator (eg, registered kinesiologist) from a local community partner organisation (hereafter referred to as the intervention team). Table 1 summarises the core components of the ACHRU-CPP. Due to the COVID-19 pandemic, some participants received virtual visits by phone or videoconferencing. A comparison of results from the virtual and in-person approaches will be published in a future paper.

### Patient and public involvement
The need for the programme was originally identified by community-dwelling older adults with diabetes and other chronic conditions and their caregivers and was subsequently codesigned by older adults in collaboration with primary and community care providers and researchers.[24] In the current RCT, patient partners from the pan-Canadian steering committee were involved in reviewing research questions and advising the research team on the selection of outcome measures.[28] Patient and public research partners also participated in local community advisory boards in each site to inform further adaptations to the intervention and support local implementation. Patient and public research partners from the local community advisory boards (RB, GG, LG, CL, DL and AM) and the steering committee (LM and FT) also provided input into development of this manuscript by reviewing and interpreting the results and helping to shape the key messages.

### Setting
The study was conducted in two sites in Ontario, Quebec and Prince Edward Island, Canada. Each of the six sites was selected to ensure variability in geographical setting (urban and rural); sociodemographic and cultural

**Table 1** The ACHRU-CPP five core components

| Intervention components | Goals |
| --- | --- |
| 1. Home/virtual visits (up to 3 home visits) and unlimited follow-up phone calls by an RN and/or RD or nutritionist | To assess older adults' and caregivers' needs and goals using standardised tools to support a coordinated care plan |
| 2. Monthly group wellness sessions (up to 6 sessions) at a local community centre led by the RN, RD or nutritionist and community programme coordinator | To provide older adults and caregivers with gentle progressive physical activity, self-management education for diabetes and other chronic conditions, and healthy lunches and snacks |
| 3. Monthly team case conferences which include an RN, RD or nutritionist and community programme coordinator | To discuss the health and social care needs of older adults and caregivers, develop and revise the coordinated care plan, and plan topics for group wellness sessions |
| 4. Collaboration with the primary care interprofessional team and other specialists (eg, family physicians, nurse practitioners, kinesiologists, social workers, home care and social service providers, pharmacists, endocrinologists) | To support primary care and community providers in working collaboratively to develop care plans for older adults, and connect older adults and caregivers to specialists and community resources |
| 5. Nurse-led care coordination/system navigation | To facilitate linkages to other primary healthcare providers, specialists and community care services for older adults and caregivers |

ACHRU-CPP, Aging, Community and Health Research Unit-Community Partnership Program; RD, registered dietitian; RN, registered nurse.

backgrounds; language spoken (English or French); demonstrated support for the ACHRU-CPP; and the presence of staff to support intervention implementation. An RN and RD or nutritionist from a primary care setting or diabetes education programme worked in partnership with a programme coordinator from a local community partner site (eg, YMCA) to implement the programme.

## Sample and recruitment

Older adults were screened for eligibility to participate in the RCT by a trained staff member of the primary care site, as described in the study protocol.[28] Eligible patients met the following inclusion criteria: (A) aged 65 years or older; (B) diagnosis of Type 1 or Type 2 diabetes with at least one other chronic condition; (C) receiving primary care services from one of the participating primary care settings; (D) living within the area served by the primary care setting and community site; (E) able to provide informed consent or has a substitute decision-maker able to provide informed consent on the patient's behalf; and (F) competent in English or French, or has an interpreter competent in English or French.

Following the completion of baseline interviews, patients were randomised to receive the intervention (ie, ACHRU-CPP) in addition to usual care or usual care alone. A total of 8–10 older adults per site who completed the 6-month intervention were invited to participate in follow-up telephone interviews. Trained research assistants (MLY and RC), with no prior relationship with participants, used a telephone script to call selected older adults within 2 weeks of completing the ACHRU-CPP, to invite them to participate in a telephone interview. Maximum variation purposive sampling[31] was used to select a diverse sample of participants across all six sites based on their sex, annual income, ethnicity and level of participation in all components of the study.

## Data collection

Semistructured postintervention telephone interviews were conducted between April 2020 and August 2021. Trained research assistants conducted audiorecorded interviews, ranging from 20 to 60 min in length, in English or French. Interviews were transcribed verbatim by experienced transcriptionists. Interviews conducted in English were transcribed and cleaned by trained research staff, while interviews conducted in French were transcribed and translated into English by professional transcriptionists and later validated by a bilingual member of the research team. Transcripts were not returned to participants for their review. The interview guide was created based on: (A) a review of the literature of health and social needs of older adults and caregivers, patient–provider communication, and system navigation and (B) feedback from patient partners and the research team with expertise in ageing, community-based supports for older adults and caregivers, and qualitative research. Box 1 provides sample interview questions.

## Data analysis

Themes were generated using the Braun and Clarke's experiential thematic analysis framework[32] and organised under relevant constructs of the Consolidated Framework for Implementation Research.[33] Thematic analysis was selected to ensure that the development of themes was informed by the experiences and perceived intervention benefits of older adults. The six phases of thematic analysis include: (A) becoming familiar with the data; (B) coding; (C) developing themes; (D) reviewing themes; (E) constructing a definition for themes and labelling them; and (F) creating a report.[32] A female research assistant with doctoral-level training in qualitative research (MLY) used the data management software NVivo V.12[34] to perform coding. MLY is fluent in both languages and

**Box 1  Sample interview questions for older adults**

Questions for older adults
1. What did you need the most in the past 6 months (eg, physical, emotional, mental or psychological support, transportation, financial assistance, housekeeping, personal care support)?
2. When [name of nurse and dietitian] visited you what types of things did they do during those visits?
3. For other people who are living with diabetes and other chronic conditions, would you recommend that a nurse or dietitian, such as [name of nurse and dietitian], visit the person at home, make phone calls or both?
4. What types of things did you do at the monthly wellness sessions or during the individual calls with [name of the community program coordinator]?
5. How, if at all, did [name of nurse and dietitian] involve you in decisions about your care?
6. How, if at all, did [name of nurse and dietitian] help you to connect with other community health or social services to help you?
7. To what extent did the nurse and dietitian help to address your needs or the issues that were most important to you?
8. How happy are you with the overall care that you received from [name of nurse and dietitian]?
9. Was the information given by [name of nurse and dietitian] and other health professionals about care consistent (across individuals)?
10. Is there anything else about your experiences with [name of nurse and dietitian] that you would like to add that we haven't already discussed?

coded in English. Results were shared with the team in English only. Following the creation of codes, these were further examined for patterns to generate themes. Themes were shared with the research team, including patient partners, to ensure they were reflective of the data.

## Rigour and trustworthiness

Consensus was reached by all authors prior to the inclusion of themes in the final report. Lincoln and Guba's validation criteria[35] were applied in this study to enhance the study's rigour. To support the credibility of findings, investigator triangulation was used in data analysis through team meetings with 5–7 members to review the coding structure and evidence of themes. These members included patient and public research partners and researchers of various disciplines with expertise in qualitative research, gerontology and community-based interventions. Conflicts were resolved through team consensus. To facilitate transferability of findings, the study sample and setting were described in detail. To support dependability and confirmability of findings, the research team kept an audit trail of study processes.

## RESULTS

A total of 295 older adults were enrolled in the RCT and randomly allocated to receive the ACHRU-CPP or usual care. At the time of data collection, 53 older adults who had completed the 6-month intervention were approached to

participate in the qualitative interviews and 45 accepted (84.9%). The rate of acceptance by site was as follows: site 1, 100% (10/10); site 2, 89.9% (8/9); site 3, 89.9% (8/9); site 4, 60% (6/10); site 5, 100% (8/8) and site 6, 71.4% (5/7). Out of the 45 participants, there was good uptake of home visits with a mean of 3.1 (SD=1.5) and group wellness sessions with a mean of 2.7 (SD=1.9). Where the providers deemed clinically necessary participants received more than the allotted three home visits. Competing commitments such as doctor appointments and lack of interest were barriers to attending group wellness sessions for some participants.

## Demographic characteristics

The mean age of the 45 older adults who participated in interviews was 71.1 years and the mean length of time living with diabetes was 18.8 years (SD=10.6). Most were female (55.6%), retired from paid work (80%), had type 2 diabetes (93.3%) and reported 4–6 chronic conditions (44.4%). Hypertension, hyperlipidaemia, and osteoarthritis (OA) and other arthritis (eg, rheumatoid arthritis) were the most reported chronic conditions. Table 2 summarises demographic characteristics of participants.

## Themes

Themes were grouped into two categories, experiences and perceived impacts of the ACHRU-CPP. Box 2 provides an overview of themes. The words in italics that label the theme are taken verbatim from transcripts. Similarly, participant quotes in the narrative that follows are noted in italics and identified by OA for older adult, # for site number and ### for participant number.

### Experiences with the ACHRU-CPP

Overall, older adults reported positive experiences with the ACHRU-CPP. They experienced: (A) in-depth dialogue with '*professional friends*'; (B) socialised with '*people with the same type of health problems*'; (C) person-centred care by '*more than one knowledgeable person*'; (D) ongoing contact with providers so '*you are not alone*'; and identified the (E) need to address ethnic/cultural differences through a '*personal session*'.

### In-depth dialogue with '*professional friends*'

In-person home and virtual visits were perceived by older adults as more relaxed compared with clinic visits and provided opportunities for in-depth dialogue about health and social issues with providers. '*The home visits are more relaxed, if you were at a clinic, you got a time slot you got to meet whatever is transacted in that timeframe*' (OA_1_152). Findings revealed that older adults had a lot of concerns and questions regarding diabetes and other chronic conditions that were often left unanswered by providers due to the nature of clinic visits that are limited by time and ailment. Home visits helped to build trust between older adults and the providers, which facilitated the exploration of concerns and needs beyond diabetes such as safe housing and transportation issues. '*I felt I could trust her [nutritionist]*' (OA_5_037). Providers were approachable

**Table 2** Demographic characteristics of interview participants

| Older adults (n=45) | |
|---|---|
| **Category** | **n (%)** |
| Age (mean (SD)): 71.7 (6.5) | |
| 65–70 | 26 (57.8) |
| 71–75 | 7 (15.6) |
| 76+ | 12 (26.7) |
| Sex | |
| Female | 25 (55.6) |
| Male | 20 (44.4) |
| Marital status | |
| Married or living with a partner | 21 (46.7) |
| Divorced, never married, separated or widowed | 23 (51.1) |
| Refused | 1 (2.2) |
| Highest level of education | |
| Completed a graduate or professional degree | 6 (13.3) |
| Completed a bachelor's degree | 10 (22.2) |
| Had some university education or completed a community college, technical college or postsecondary programme | 12 (26.7) |
| Completed secondary school | 10 (22.2) |
| Did not complete secondary school | 7 (15.6) |
| Current employment status | |
| Retired from paid work | 36 (80.0) |
| Employed full time | 4 (8.9) |
| Employed part time | 2 (4.4) |
| Unemployed and looking for work | 1 (2.2) |
| Refused | 2 (4.4) |
| Annual household income | |
| CAD$150 000 or more | 2 (4.4) |
| CAD$100 000 or more, but less than CAD$150 000 | 2 (4.4) |
| CAD$50 000 or more, but less than CAD$100 000 | 12 (26.7) |
| CAD$20 000 or more, but less than CAD$50 000 | 16 (35.6) |
| Less than CAD$20 000 | 11 (24.4) |
| Refused | 2 (4.4) |
| Born in Canada | |
| Yes | 31 (68.9) |
| Ethnic/racial group | |
| White/Caucasian | 32 (71.1) |
| Black | 3 (6.7) |
| Caribbean/Guyanese | 3 (6.7) |
| Filipino | 2 (4.4) |
| First Nations | 1 (2.2) |
| South Asian | 1 (2.2) |
| Southeast Asian | 1 (2.2) |
| Chinese | 1 (2.2) |
| Japanese | 1 (2.2) |
| Language(s) Spoken | |

**Table 2** Continued

| Older adults (n=45) | |
|---|---|
| **Category** | **n (%)** |
| English | 37 (82.2) |
| French | 15 (33.3) |
| Living with others (eg, spouse, children, other relative, friend, group setting) | |
| Yes | 27 (60.0) |
| Type of diabetes | |
| Type 1 diabetes | 1 (2.2) |
| Type 2 diabetes | 42 (93.3) |
| Unknown | 2 (4.4) |
| No. of chronic conditions (mean (SD)): 5.6(2.9) | |
| 1–3 | 11 (24.4) |
| 4–6 | 20 (44.4) |
| 7–9 | 8 (17.8) |
| 10 + | 6 (13.3) |
| Commonly reported chronic conditions | |
| Hypertension | 34 (75.6) |
| Hyperlipidaemia | 27 (60.0) |
| Osteoarthritis and other arthritis | 18 (40.0) |
| Cardiovascular disease | 16 (35.6) |
| At least 1 emergency room visit in the last 6 months | |
| 6 months prior to ACHRU-CPP | 8 (17.8) |
| 6-month follow-up | 7 (15.6) |

ACHRU-CPP, Aging, Community and Health Research Unit-Community Partnership Program.

and understanding of older adults' situations and were considered as friends and confidants. '*They [providers] were professional friends*' (OA_2_242). Their approach was especially important when discussing sensitive topics such as mental health concerns. '*I had my sick niece calling me and that was stressing me, so she [nurse] said to 'Let go. When we*

---

**Box 2    Themes of older adult experiences and perceived impacts with the Aging, Community and Health Research Unit-Community Partnership Program (ACHRU-CPP)**

Experiences with the ACHRU-CPP
⇒ In-depth dialogue with '*professional friends*'.
⇒ Socialised with '*people with the same type of health problems*'.
⇒ Person-centred care by '*more than one knowledgeable person*'.
⇒ Ongoing contact with providers so '*you are not alone*'.
⇒ Need to address ethnic/cultural differences through a '*personal session*'.
Perceived Impacts of the ACHRU-CPP
⇒ Improved diabetes self-management behaviours: '*make more proactive steps*'.
⇒ Added connection to health and social support services '*that could help me*'.

*are stressed, that's not always good'. So how to manage my stress'* (OA_6_023).

### Socialised with '*people with the same type of health problems'*

Older adults and caregivers perceived that group wellness sessions helped them meet others who understood what it is like to live with diabetes and other chronic conditions. '*The fact of socializing with other people with the same types of health problems as we do'* (OA_5_037). Learning about the burden that others experience with their conditions encouraged older adults to support each other and express their frustrations. The sessions provided opportunities for group exercises which provided peer motivation. The group sessions were particularly helpful for older adults who were socially isolated, and some older adults became friends because of the sessions.

### Person-centred care by '*more than one knowledgeable person'*

Older adults appreciated that they received person-centred care from a team of providers through the ACHRU-CPP to discuss diabetes, their other chronic conditions and social concerns. Some older adults perceived that in usual care providers at times delivered care in silos or independent of other disciplines. Older adults valued providers working collaboratively to meet their needs. '*It was good that they worked in a team. More than one knowledgeable person. That was important'* (OA_2_242). They felt that providers were listening to their concerns and that, prior to meeting with the intervention team, it was difficult to find the right person to talk to about diabetes.

> I enjoyed having them come to visit. I don't talk to a lot of people about my diabetes because I don't feel it's that complicated, but nobody really wants to listen about your health issues.(OA_3_032)

Providers supported the management of other conditions in addition to diabetes. '*I was having troubles with my bowels, but we got that regulated and it's good'* (OA_4_075).

### Ongoing contact with providers so '*you are not alone'.*

Providers made older adults feel that someone was concerned about their well-being. '*It's not as if we are just left alone with our problems. What you are doing is very good; continue'* (OA_5_128). Managing diabetes and MCC felt burdensome for older adults, and they appreciated regular contact with providers. Follow-up phone calls were well received by older adults, especially by those living alone or with little support, and ensured that they '*haven't fallen through the cracks'* (OA_3_058). The ongoing follow-up with the team reinforced familiar information that older adults had forgotten to put into practice over time. A few participants reported that once the intervention stopped at the 6-month period they felt that there was a break in the social connection with peers and formal providers.

### Need to address ethnic/cultural differences through a '*personal session'*

Some ethnic groups may have language barriers and be '*very shy and they don't approach people unless someone else pushes them to go [join programs]'* (OA_1_061). Some older adults perceived that individual wellness sessions with providers may be helpful for those with language barriers. '*Some of them had a bit of a language problem. I think a personal session would be much more helpful'* (OA_2_086). The following challenge experienced by one participant also exemplifies the need to provide dedicated time, encouragement and support for older adults to share their personal cultural practices during interactions with peers and providers.

> One of the things I learned about myself from my community [Indigenous community] and my family is that eating has to do with seasons…Your year-round diet has to do with what's available to you…I mentioned that one time in the group [Group Wellness Sessions]and they thought that had nothing to do with what the topics were. (OA_2_013)

Providing opportunities to discuss cultural practices promotes diversity within groups and learning across cultures.

### Perceived impacts

Older adults perceived that the ACHRU-CPP had positive impacts on their health and well-being as a result of: (A) improved diabetes self-management behaviours: '*make more proactive steps'* and (B) added connection to health and social support services '*that could help me'.*

### Improved diabetes self-management behaviours: '*make more proactive steps'*

Older adults felt that the ACHRU-CPP helped them to recognise and take more action in preventing complications, such as hypoglycaemia and hyperglycaemia, and decreasing their blood pressure, as described in the following quote. They recognised that reaching a level of effective self-management of chronic conditions can be complicated and they appreciated the support provided through the ACHRU-CPP.

> What [the nurse] and [dietitian] caused me to be concerned about is to make more proactive steps, to watch out for those low blood sugars. I really didn't realize how badly they could affect you. Shaking and double vision is one thing but not being able to drive, that's quite another thing. (OA_2_242)

Older adults indicated that they gained nutrition knowledge by participating in the ACHRU-CPP. '*I improved it [eating habits]. I had to eat more fruit and vegetables…and after that, I had to hydrate myself more and add more fibre to my diet'* (OA_5_027). Older adults perceived that changes made to their diet could lead to multiple benefits including weight loss and decreased sugar levels. '…*drinking more water and diet, I think that's what was important, and I lost*

*weight at the same time. By eating well, fewer treats, being more careful, the sugar levels were lower'* (OA_6_005).

Older adults perceived they were able to build more muscle mass and lose weight and experienced less difficulty in climbing stairs. Some older adults felt that they were not exercising enough prior to participating in the ACHRU-CPP and perceived that the providers helped them to meet their activity goals.

They really helped me with the exercise piece. I had poor balance. It was the [nurse and dietitian] that really said 'why don't you try doing this? I'll give you a call this week and see if you got out to do your walk', and then I'd promise them that I would start journaling my steps so little by little I started increasing my exercise. (OA_2_247)

### Added connection to health and social support services '*that could help me'*

Older adults indicated that they were referred to and connected with health and social support services (eg, food bank, exercise programme, smoking cessation, home care, social work, arts programme). Living well with diabetes and other chronic conditions was perceived by older adults to require more than just medical care. Some older adults required supports to meet their basic needs, as they were not able to afford groceries or travel far distances for groceries and medications. '*She [nurse] referred me [for medication delivery]'* (OA_1_061). Some older adults required mental health support to enhance their ability to manage diabetes and other chronic conditions. The intervention team followed-up with older adults after making referrals to ensure that they were connected. Older adults were referred to local community resources that offered free or low-cost services. '*They told me I could go to [name of community centre] to do exercise'* (OA_4_016). Prior to participating in the ACHRU-CPP older adults indicated that they had seldom been referred to programmes outside of the clinic, and therefore, they were not aware of available community resources.

Older adults found it important to be aware of publicly funded resources, such as tax rebates and housing options, in case they or their loved one required these in the future. By attending group wellness sessions, they learnt about the types of programmes the community partner sites had to offer. '*They [community partner site] have virtual classes and they're all free. They have special classes just for seniors'* (OA_2_013). Despite the mostly positive impacts of being connected with relevant health and social services, some participants reported that not all of their needs were met at the group wellness sessions. For example, some did not qualify for certain financial assistance programmes. '*They just said you have to qualify [for financial assistance] because…you have to apply for this, you have to apply for this'* (OA_1_058). Some participants reported that it was difficult to absorb all of the information presented and not all of the information was relevant to their situation. It was challenging to tailor group wellness sessions to individual needs compared with home visits which allowed further individualised tailoring.

## DISCUSSION

Key findings of this study were that the ACHRU-CPP increased in-depth dialogue with '*professional friends'* and provided person-centred care and ongoing contact with providers to prevent feelings of being alone. The use of a social determinants of health approach by the intervention teams was a novel component of the programme and highlighted the importance of addressing social aspects of care for older adults with diabetes and MCC. Group interactions brought together participants with the same type of health issues and provided peer motivation and support. Participants identified that the programme would benefit from adaptations to address cultural and language differences among older adults living with diabetes and other conditions in Canada.

Older adults with multimorbidity and diabetes face significant burden in managing their chronic conditions, which challenge their self-care and adversely affect their overall quality of life.[36] In addition to managing diabetes symptoms and complications (eg, hypoglycaemia and hyperglycaemia, visual impairment, neuropathy) which negatively impact their physical functioning, older adults with diabetes often experience psychological burden associated with complex medication and diet regimens (eg, medication and insulin management and glycaemic control).[36] Added pressures include financial burden, food insecurity, social isolation, lack of social support, as well as frailty, and the burden of comorbid chronic conditions, for example, anxiety, depression.[36] Several of these challenges were experienced by our study participants. The largely positive response to the programme could be attributed to the support that participants received in managing the burden associated with living with diabetes and other chronic conditions. High-quality care for this population to prevent diabetes distress (ie, challenges faced when dealing with the demands of diabetes) requires good communication and trusting relationships with providers, social and peer support, and self-management education.[37]

A novel finding of this study was that the ACHRU-CPP was perceived by older adults from three Canadian provinces to positively impact their self-management practices of diabetes and MCC by helping to address their broad health and social needs. This has not been documented before in similar studies.[19 20] This may be because the ACHRU-CPP was longer and more person-centred compared with other interventions. Interventionists were able to directly assess the home context and understand the impacts of social determinants of health. Mental health concerns and lack of support, which were experienced by our study participants, can impact the ability of older adults to effectively manage diabetes and lead to severe hypoglycaemia, elevated HbA1c levels, a greater

number of missed insulin doses, and a higher risk for diabetic ketoacidosis and mortality.[3 38 39]

When healthcare providers recommend lifestyle changes, they need to recognise that social determinants of health such as housing, food security, social relationships and financial stability have an impact on older adults' abilities to enact them.[40] In the current study, the intervention team assessed the social determinants of health and found ways to address them, such as by linking older adults with relevant community resources, to help overcome barriers to self-management. The intervention team targeted health literacy of older adults and caregivers through education, capacity building, and opportunities for dialogue among peers and experts.

In the current study, older adults appreciated receiving person-centred care supported by a team of providers from primary care and community sectors and the engagement of providers outside of the intervention team (such as social workers). Due to the complex nature of diabetes and MCC, interprofessional collaboration has been found to lead to positive outcomes for persons with type 2 diabetes, such as improvements in HbA1c levels, regular testing of blood glucose levels and smoking cessation.[41 42]

What is unique about this study is the partnership between healthcare providers and a programme coordinator from a local community partner site. These health and social services can be underused if healthcare providers are not aware of them. As per the Quintuple Aim,[23] there is a need to optimise the use of existing community-based services for patients, address any barriers to accessing these services and for strong coordination of services.[43]

Person-centred care was perceived to be key strength of the ACHRU-CPP that enabled older adults to improve self-management practices related to diabetes and MCC. In working towards a person-centred learning health system, defined as a health system that integrates internal data, patient experience and research evidence,[44] the priorities and experiences of older adults should be regularly reported in data systems so that services that meet their needs are developed and evaluated as part of continuous quality improvement processes.[38] As seen in this study, patient experience can be improved by having a provider connect patients with other interdisciplinary health and social care providers to ensure that smooth transitions between services occur.[45]

In practice and policy, there is a need for integrated care delivery models that leverage community partnerships to help fill gaps in meeting the complex health and social needs of older adults with diabetes. To advance Quintuple Aim outcomes,[23] it is critical to assess patient experiences with healthcare services as part of intervention research and practice to improve health system performance.

The strengths of the study include its rigorous qualitative design and large sample size, and the inclusion of diverse participants with regard to sex, marital status and annual income, from multiple sites across Canada. A limitation of the study was related to the sample as there was a lack of cultural diversity and under-representation of older adults from marginalised communities. A Diabetes Canada roundtable of key stakeholders recently emphasised the need to implement community-based interventions,[46] such as the ACHRU-CPP, to better support older adults with diabetes and MCC living in marginalised communities. The need for community-based interventions is based on the premise that racial and socioeconomic disparities disproportionally affect marginalised older adults with diabetes and MCC and put them at an increased risk for diabetes complications and mortality.[47]

## CONCLUSION

Overall, older adults with diabetes and MCC reported a positive experience and felt that the ACHRU-CPP had a positive impact in supporting diabetes self-management. Study findings reveal the need to ensure that older adults receive ongoing support and contact with a collaborative team of primary care and community providers to better meet the complex needs associated with daily self-management of diabetes and MCC. Results also shine light on the broader social context that constitutes the life world of older adults and how chronic disease self-management interventions need to address these contexts comprehensively through tailoring to individual circumstances. It is our hope that these findings will help usher in a new era of contextually informed person-centred care.

**Author affiliations**

[1]School of Nursing, McMaster University Faculty of Health Sciences, Hamilton, Ontario, Canada

[2]School of Nursing, Aging, Community and Health Research Unit, McMaster University Faculty of Health Sciences, Hamilton, Ontario, Canada

[3]Department of Applied Human Sciences, Faculty of Science, University of Prince Edward Island, Charlottetown, Prince Edward Island, Canada

[4]VITAM Centre de recherche en santé durable, Université Laval, Quebec, Quebec, Canada

[5]Department of Family and Emergency Medicine, Faculty of Medicine, Université Laval, Quebec, Quebec, Canada

**Acknowledgements** We thank the older adults and caregivers who participated in this study, as well as the nurses, dietitians, nutritionists and community programme coordinators who provided the intervention. We also thank the managers of intervention teams, the recruiters, research assistants and the study sites for their support of this study. Thanks to Robyn Connors (RC) for conducting interviews with study participants. We thank the research team in the Aging, Community and Health Research Unit (https://achru.mcmaster.ca/) for supporting this study. We would also like to thank the Community Advisory Board members, including people with lived experience with diabetes, who supported local implementation of the project in each province. We acknowledge the following authors who are members of the ACHRU-CPP Research Team: Jenny Ploeg, School of Nursing, Faculty of Health Sciences, McMaster University, Hamilton, Ontario, Canada; Maureen Markle-Reid, School of Nursing, Faculty of Health Sciences, McMaster University, Hamilton, Ontario, Canada; Ruta Valaitis, School of Nursing, Faculty of Health Sciences, McMaster University, Hamilton, Ontario, Canada; Kathryn Fisher, School of Nursing, Aging, Community and Health Research Unit, Faculty of Health Sciences, McMaster University, Hamilton, Ontario, Canada; Rebecca Ganann, School of Nursing, Aging,

Community and Health Research Unit, Faculty of Health Sciences, McMaster University, Hamilton, Ontario, Canada; Johanne Blais, Department of Family Medicine and Emergency Medicine, Faculty of Medicine, Université Laval, Quebec City, Quebec; Andrea Gruneir, Department of Family Medicine Research Program, University of Alberta, Edmonton, Alberta, Canada; France Légaré, VITAM-Centre de recherche en santé durable, Université Laval, Quebec City, Quebec, Canada; Janet MacIntyre, Faculty of Nursing, University of Prince Edward Island, Charlottetown, Prince Edward Island, Canada; William Montelpare, Department of Applied Human Sciences, Faculty of Science, University of Prince Edward Island, Prince Edward Island, Canada; Jean-Sébastien Paquette, Department of Family Medicine and Emergency Medicine, Faculty of Medicine Université Laval, Québec, Canada; Marie-Eve Poitras, Department of Family Medicine and Emergency Medicine, Faculty of Medicine and Health Sciences, Université de Sherbrooke Chicoutimi, Quebec, Canada; Angela Riveroll, Department of Applied Human Sciences, Faculty of Science, University of Prince Edward Island, Charlottetown, Prince Edward Island, Canada, Ali Ben Charif, CubecXpert, Quebec City, Quebec, Canada; Dean Eurich, School of Public Health, University of Alberta, Edmonton, Alberta, Canada; Amiram Gafni, Department of Health Research Methods, Evidence and Impact, Faculty of Health Sciences, McMaster University, Hamilton, Ontario, Canada; Gary Lewis, Department of Medicine and Department of Physiology, University of Toronto, Toronto, Ontario, Canada; Lynne Mansell, Patient/Public Research Partner, Alberta, Canada; Melissa Northwood, School of Nursing, Aging, Community and Health Research Unit, Faculty of Health Sciences, McMaster University, Hamilton, Ontario, Canada; Janet Pritchard, Interdisciplinary Science and Kinesiology, Faculty of Science, McMaster University, Hamilton, Ontario, Canada; Cheryl Sadowski, Faculty of Pharmacy and Pharmaceutical Sciences, University of Alberta, Edmonton, Alberta, Canada; Diana Sherifali, School of Nursing, Faculty of Health Sciences, McMaster University, Hamilton, Ontario, Canada; Frank Tang, Patient/Public Research Partner, Ontario, Canada; Lehana Thabane, Department of Health Research Methods, Evidence and Impact, Faculty of Health Sciences, McMaster University, Hamilton, Ontario, Canada; Ross Upshur, Bridgepoint Active Healthcare, Toronto, Ontario, Canada; Tyler Williamson, Centre for Health Informatics, Cumming School of Medicine and Department of Community Health Sciences, University of Calgary, Calgary, Alberta, Canada; Marie-Lee Yous, School of Nursing, Faculty of Health Sciences, McMaster University, Hamilton, Ontario, Canada.

**Collaborators** The ACHRU-CPP Research Team: Jenny Ploeg, School of Nursing, Faculty of Health Sciences, McMaster University, Hamilton, Ontario, Canada; Maureen Markle-Reid, School of Nursing, Faculty of Health Sciences, McMaster University, Hamilton, Ontario, Canada; Ruta Valaitis, School of Nursing, Faculty of Health Sciences, McMaster University, Hamilton, Ontario, Canada; Kathryn Fisher, School of Nursing, Aging, Community and Health Research Unit, Faculty of Health Sciences, McMaster University, Hamilton, Ontario, Canada; Rebecca Ganann, School of Nursing, Aging, Community and Health Research Unit, Faculty of Health Sciences, McMaster University, Hamilton, Ontario, Canada; Johanne Blais, Department of Family Medicine and Emergency Medicine, Faculty of Medicine, Université Laval, Quebec City, Quebec; Andrea Gruneir, Department of Family Medicine Research Program, University of Alberta, Edmonton, Alberta, Canada; France Légaré, VITAM-Centre de recherche en santé durable, Université Laval, Quebec City, Quebec, Canada; Janet MacIntyre, Faculty of Nursing, University of Prince Edward Island, Charlottetown, Prince Edward Island, Canada; William Montelpare, Department of Applied Human Sciences, Faculty of Science, University of Prince Edward Island, Prince Edward Island, Canada; Jean-Sébastien Paquette, Department of Family Medicine and Emergency Medicine, Faculty of Medicine Université Laval, Québec, Canada; Marie-Eve Poitras, Department of Family Medicine and Emergency Medicine, Faculty of Medicine and Health Sciences, Université de Sherbrooke Chicoutimi, Quebec, Canada; Angela Riveroll, Department of Applied Human Sciences, Faculty of Science, University of Prince Edward Island, Charlottetown, Prince Edward Island, Canada, Ali Ben Charif, CubecXpert, Quebec City, Quebec, Canada; Dean Eurich, School of Public Health, University of Alberta, Edmonton, Alberta, Canada; Amiram Gafni, Department of Health Research Methods, Evidence and Impact, Faculty of Health Sciences, McMaster University, Hamilton, Ontario, Canada; Gary Lewis, Department of Medicine and Department of Physiology, University of Toronto, Toronto, Ontario, Canada; Lynne Mansell, Patient/Public Research Partner, Alberta, Canada; Melissa Northwood, School of Nursing, Aging, Community and Health Research Unit, Faculty of Health Sciences, McMaster University, Hamilton, Ontario, Canada; Janet Pritchard, Interdisciplinary Science and Kinesiology, Faculty of Science, McMaster University, Hamilton, Ontario, Canada; Cheryl Sadowski, Faculty of Pharmacy and Pharmaceutical Sciences, University of Alberta, Edmonton, Alberta, Canada; Diana Sherifali, School of Nursing, Faculty of Health Sciences, McMaster University, Hamilton, Ontario, Canada; Frank Tang, Patient/Public Research Partner, Ontario, Canada; Lehana Thabane, Department of Health Research Methods, Evidence and Impact, Faculty of Health Sciences, McMaster University, Hamilton, Ontario, Canada; Ross Upshur, Bridgepoint Active Healthcare, Toronto, Ontario, Canada; Tyler Williamson, Centre for Health Informatics, Cumming School of Medicine and Department of Community Health Sciences, University of Calgary, Calgary, Alberta, Canada; Marie-Lee Yous, School of Nursing, Faculty of Health Sciences, McMaster University, Hamilton, Ontario, Canada.

**Contributors** Conceptualisation: JP, MMR, RV, KF, RG, FL and WM. Formal analysis: MLY, RG, TC and JP. Funding acquisition: JP, MMR, RV, KF, RG, AG, FL, JM and WM. Investigation: JP, MMR, RV, KF, RG, TC, FL, WM and MLY. Methodology: JP, MMR, RV, KF, RG, MLY, FL and WM. Project administration: JP, MMR, RV, TC, FL and WM. Resources: JP, MMR, RV, FL and WM. Supervision: JP, MMR, RV, RG, TC, FL and WM. Validation: MLY, JP, MMR, RV, KF, RG, TC and RC. Writing—original draft: MLY. Writing—review and editing: MLY, RG, TC, JP, MMR, RV, KF, FL, JM and WM. Guarantor: MMR. The authors read and approved the final manuscript.

**Funding** This study is supported, in part, by funding from the Canadian Institutes of Health Research Strategy for Patient-Oriented Research (SPOR) Primary and Integrated Health Care Innovations Network: Programmatic Grants (Funding Reference Number: KPG-156883) in partnership with: Diabetes Action Canada, a Canadian Institutes for Health Research (CIHR) Strategy for Patient-Oriented Research Network in Chronic Disease (project reference #1.1.1ACHR); McMaster Institute for Research on Aging (Hamilton, ON); McMaster University School of Nursing; Réseau-1 Québec; Fonds de Recherche du Québec (FRQS); Scarborough Health Network Foundation. This research was also undertaken, in part, thanks to the funding from Dr. Markle-Reid's Tier 2 CIHR Canada Research Chair. The study was investigator initiated.

**Disclaimer** The funders of this study had no role in study design, data collection, data analysis, data interpretation or writing the manuscript.

**Competing interests** None declared.

**Patient and public involvement** Patients and/or the public were involved in the design, or conduct, or reporting, or dissemination plans of this research. Refer to the Methods section for further details.

**Patient consent for publication** Not applicable.

**Ethics approval** Institutional ethics approval was obtained from the following: the Hamilton Integrated Research Ethics Board (#5101); the Scarborough Health Network Research Ethics Board (#NEP-18-014); the Unity Health Toronto Research Ethics Board (#18-336); University of Prince Edward Island Research Ethics Board (#6008019); Prince Edward Island Research Ethics Board (#6008019); and Centre intégré universitaire de santé et de services sociaux (CIUSSS) de la Capitale-Nationale (MP-13-2019-1670). Participants gave informed consent to participate in the study before taking part.

**Provenance and peer review** Not commissioned; externally peer reviewed.

**Data availability statement** No data are available. The data for this research consists of questionnaires and interviews. Raw data such as audiofiles and interview transcripts cannot be publicly released due to the risk of compromising participant confidentiality related to identification of voices and publicly exposing personal information.

**ORCID iDs**
Marie-Lee Yous http://orcid.org/0000-0002-4271-0401
Rebecca Ganann http://orcid.org/0000-0002-7566-8932
Jenny Ploeg http://orcid.org/0000-0001-8168-8449
Melissa Northwood http://orcid.org/0000-0001-5043-8068
Ruta Valaitis http://orcid.org/0000-0002-3117-0542
Tracey Chambers http://orcid.org/0000-0002-9325-6894
France Légaré http://orcid.org/0000-0002-2296-6696

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
