## [Reviewer comments · BMJ Open]

ARTICLE DETAILS

TITLE (PROVISIONAL)	Older Adults' Experiences and Perceived Impacts of the Aging, Community and Health Research Unit – Community Partnership Program (ACHRU-CPP) for Diabetes Self-Management in Canada: A Qualitative Descriptive Study
AUTHORS	Yous, Marie-Lee; Ganann, Rebecca; Ploeg, Jenny; Markle-Reid, Maureen; Northwood, Melissa; Fisher, Kathryn; Valaitis, Ruta; Chambers, Tracey; Montelpare, William; Légaré, France; Beleno, Ron; Gaudet, Gary; Giacometti, Luisa; Levely, Deborah; Lindsay, Craig; Morrison, Allan; Tang, Frank; ACHRU-CPP Research Team, on behalf of the ACHRU-CPP Research Team

VERSION 1 – REVIEW

REVIEWER	Ogrin, Rajna Bolton Clarke Support Centre, Bolton Clarke Research Institute
REVIEW RETURNED	13-Oct-2022

GENERAL COMMENTS	Very well written article of a project that is well considered and actually developed to meet the needs of community members. Great work. There isn't a great deal I can suggest to improve this paper. Some thoughts:  1. Suggest authors include an average age of participants and duration of diabetes in the abstract; 2. Results - I think it is important to include information around participants engagement in the program. This is a Hybrid Type II implementation design - so this information would be available. It is important to know whether the experiences shared by participant relates to their level of participation in the program. 3. Results - the data is skewed to positive findings. I am sorry, but having worked with people living with diabetes for >20 years, one thing I am very aware of is the incredible burden many individuals feel as a result of their condition. And the management burden is significant. However, while I can't imagine there were only positive experiences, the questions asked are skewed towards encouraging positive responses. I would suggest including some realistic information around issues raised would provide more balanced reporting. 4. Discussion - including some literature on issues related to this care would be of value. 5. Discussion - no mention of limitations. Overall, a great paper and I look forward to reading the other components.
--

REVIEWER	Kalula, Sebastiana
-----------------	--------------------

	Faculty of Health Science, university of Cape Town, South Africa, Department of Medicine Cape Town , Western Cape, South Africa, Geriatric Medicine
REVIEW RETURNED	09-Dec-2022

GENERAL COMMENTS	The authors are to be commended for the clear presentation of the conduct of this study and a clearly written manuscript. The only points I have that require some clarity are:  1. How many participants did they recruit from each of the six sites and how many of these did agree to the follow-up interview? 2. The data were de-identified using unique IDs, but the authors indicate that the data is not made available due to possible identification of participants. Is this identification of their voices? 3. Line 147: The YMCA is not written in full anywhere in the manuscript..
---

VERSION 1 – AUTHOR RESPONSE

Reviewer: 1

Dr. Rajna Ogrin, Bolton Clarke Support Centre

Comments to the Author:

Very well written article of a project that is well considered and actually developed to meet the needs of community members. Great work. There isn't a great deal I can suggest to improve this paper.

Some thoughts:

1. Suggest authors include an average age of participants and duration of diabetes in the abstract;

Response: Thanks very much for the feedback and excellent suggestions. We have now included in the abstract and the paper the average age of participants and duration of diabetes:

“The mean age of the 45 older adults who participated in interviews was 71.1 years and the mean length of time living with diabetes was 18.8 years (SD (standard deviation) = 10.6).”

2. Results - I think it is important to include information around participants engagement in the program. This is a Hybrid Type II implementation design - so this information would be available. It is important to know whether the experiences shared by participant relates to their level of participation in the program.

Response:

We have included the following in the “Results”:

“Out of the 45 participants, there was good uptake of home visits with a mean of 3.1 (SD (standard deviation) = 1.5) and group wellness sessions with a mean of 2.7 (SD = 1.9). Where the providers deemed clinically necessary participants received more than the allotted 3 home visits. Competing commitments such as doctor appointments and lack of interest were barriers to attending group wellness sessions for some participants.”

3. Results - the data is skewed to positive findings. I am sorry, but having worked with people living with diabetes for >20 years, one thing I am very aware of is the incredible burden many individuals feel as a result of their condition. And the management burden is significant. However, while I can't imagine there were only positive experiences, the questions asked are skewed towards encouraging

positive responses. I would suggest including some realistic information around issues raised would provide more balanced reporting.

Response:

Thank you for your insights. We have made additions throughout the 'Results' section to illustrate the complexity and burden associated with diabetes and other chronic conditions. The fact that older adults responded so positively to the intervention reveals that a lot of their care needs were not being met in usual care.

For example, we included the following:

"Despite the mostly positive impacts of being connected with relevant health and social services, some participants reported that not all of their needs were met at the group wellness sessions. For example, some did not qualify for certain financial assistance programs. "They just said you have to qualify [for financial assistance] because...you have to apply for this, you have to apply for this" (OA_1_058). Some participants reported that it was difficult to absorb all of the information presented and not all of the information was relevant to their situation. It was challenging to tailor group wellness sessions to individual needs compared to home visits which allowed further individualized tailoring."

4. Discussion - including some literature on issues related to this care would be of value.

Response:

We have included the following in the 'Discussion' section.

"Older adults with multimorbidity and diabetes face significant burden in managing their chronic conditions, which challenge their self-care and adversely affect their overall quality of life [36]. In addition to managing diabetes symptoms and complications (e.g., hypoglycemia and hyperglycemia, visual impairment, neuropathy) which negatively impact their physical functioning, older adults with diabetes often experience psychological burden associated with complex medication and diet regimens (e.g., medication and insulin management and glycemic control) [36]. Added pressures include financial burden, food insecurity, social isolation, lack of social support, as well as frailty, and the burden of comorbid chronic conditions, e.g., anxiety, depression [36]. Several of these challenges were experienced by our study participants. The largely positive response to the program could be attributed to the support that participants received in managing the burden associated with living with diabetes and other chronic conditions. High quality care for this population to prevent diabetes distress (i.e., challenges faced when dealing with the demands of diabetes) requires good communication and trusting relationships with providers, social and peer support, and self-management education [37]."

5. Discussion - no mention of limitations.

Response:

In the last paragraph of the 'Discussion' section we have included the following limitation.

"A limitation of the study was related to the sample as there was a lack of cultural diversity and under-representation of older adults from marginalized communities."

Overall, a great paper and I look forward to reading the other components.

Response: Thanks very much for the feedback.

Reviewer: 2

Dr. Sebastiana Kalula, Faculty of Health Science, university of Cape Town, South Africa, Department of Medicine Cape Town, Western Cape, South Africa, University of Cape Town

Comments to the Author:

The authors are to be commended for the clear presentation of the conduct of this study and a clearly written manuscript. The only points I have that require some clarity are:

1. How many participants did they recruit from each of the six sites and how many of these did agree to the follow-up interview?

Response:

Thanks very much for your feedback and excellent suggestions. In the results section we have included the following:

“The rate of acceptance by site was as follows: Site 1, 100% (10/10); Site 2, 88.9% (8/9); Site 3, 88.9% (8/9); Site 4, 60% (6/10); Site 5, 100% (8/8); and Site 6, 71.4% (5/7).”

2. The data were de-identified using unique IDs, but the authors indicate that the data is not made available due to possible identification of participants. Is this identification of their voices?

Response:

We have revised the sentence in the ‘Availability of Data and Materials’ section to the following: Raw data such as audio-files and interview transcripts cannot be publicly released due to the risk of compromising participant confidentiality related to identification of voices and publicly exposing personal information.

3. Line 147: The YMCA is not written in full anywhere in the manuscript..

Response:

The YMCA is not written in full to be consistent with its use by YMCA Canada. From the YMCA Canada website:

“When first established, Y.M.C.A. was an acronym for Young Men’s Christian Association. While you will still find that some YMCAs around the world still use this full name, many, including all YMCAs in Canada, are known simply as the YMCA. This reflects our commitment to serving people of all ages, genders, backgrounds, and abilities through all stages of life.”

<https://www.ymca.ca/faqs#:~:text=What%20does%20'YMCA'%20stand%20for,known%20simply%20as%20the%20YMCA.>

Clean and marked copies of the revised manuscript have been uploaded to the ScholarOne submission system.

We acknowledge that the paper exceeds the 4,000-word limit (current word count is 4,672); however, this was necessary to respond to the reviewers’ comments, and strengthens the paper.

Thank you again for the reviewers’ comments and for the opportunity to make these revisions.

Sincerely,

Marie-Lee Yous, RN, MSc, PhD
Faculty of Health Sciences, School of Nursing
McMaster University
1280 Main Street West
Hamilton, Ontario, Canada
L8S 4K1
Email: yousm@mcmaster.ca
Phone: 905-920-4614

VERSION 2 – REVIEW

REVIEWER	Ogrin, Rajna Bolton Clarke Support Centre, Bolton Clarke Research Institute
REVIEW RETURNED	24-Feb-2023

GENERAL COMMENTS	Great work. A couple of minor things: RESULTS: 1. This quote is a bit odd: Line 364 - 365 There is a need to allow dedicated time for older adults to share their cultural practices during interactions with peers and the intervention team. “One of the things I learned about myself from my community [Indigenous community] and my family is that eating has to do with seasons...Your year-round diet has to do with what’s available to you...I mentioned that one time in the group [Group Wellness Sessions] and they thought that had nothing to do with what the topics were” (OA_2_013) 2. Suggest reframing – avoid the term ‘poor diabetes care’ – suggest consider utilising the Language Matters in Diabetes guidance: Our language matters: Improving communication with and about people with diabetes. A position statement by Diabetes Australia - ScienceDirect resource - regarding the following: Line 380-381: They reported that providers helped them to be alert to complications that can arise from *poor* diabetes care. DISCUSSION 1. Suggest first paragraph of discussion should also include how important those social care aspects were. This is a novel component that team healthcare generally does not include. Great work
--

VERSION 2 – AUTHOR RESPONSE

Reviewer: 1

Dr. Rajna Ogrin, Bolton Clarke Support Centre

Comments to the Author:

Great work. A couple of minor things:

RESULTS:

1. This quote is a bit odd: Line 364 - 365

There is a need to allow dedicated time for older adults to share their cultural practices during interactions with peers and the intervention team.

“One of the things I learned about myself from my community [Indigenous community] and my family is that eating has to do with seasons...Your year-round diet has to do with what’s available to you...I mentioned that one time in the group [Group Wellness Sessions] and they thought that had nothing to do with what the topics were” (OA_2_013)

Response: The inclusion of this quote was meant to illustrate the need to allow participants time to discuss cultural practices. To explain this quote, we have removed the sentence, “There is a need to

allow dedicated time for older adults to share their cultural practices during interactions with peers and the intervention team.” and added the following sentences, “The following challenge experienced by one participant also exemplifies the need to provide dedicated time, encouragement and support for older adults to share their personal cultural practices during interactions with peers and providers.” and “Providing opportunities to discuss cultural practices promotes diversity within groups and learning across cultures.

2. Suggest reframing – avoid the term ‘poor diabetes care’ – suggest consider utilising the Language Matters in Diabetes guidance: Our language matters: Improving communication with and about people with diabetes. A position statement by Diabetes Australia - ScienceDirect resource - regarding the following:

Line 380-381:

They reported that providers helped them to be alert to complications that can arise from *poor* diabetes care.

Response: Thank you for sharing the document regarding inclusive words in discussing diabetes. We have removed the sentence, “They reported that providers helped them to be alert to complications that can arise from poor diabetes care.” and included the following sentence, “Older adults felt that the ACHRU-CPP helped them to recognize and take more action in preventing complications, such as hypoglycemia and hyperglycemia, and decreasing their blood pressure, as described in the following quote.”

DISCUSSION

1. Suggest first paragraph of discussion should also include how important those social care aspects were. This is a novel component that team healthcare generally does not include.

Great work

Response: Thank you for your suggestion. We agree with the need to highlight how our intervention was novel in addressing social care aspects. We added the following sentence to the first paragraph of the discussion: “The use of a social determinants of health approach by the intervention teams was a novel component of the program and highlighted the importance of addressing social aspects of care for older adults with diabetes and MCC.”

Additional changes:

1. The sentence “Community-based interventions were found to be most effective for Type 2 diabetes self-management compared to other interventions [21].” has been removed.

2. A few minor changes have been made throughout the text to correct grammar and spelling and improve clarity.

Clean and marked copies of the revised manuscript have been uploaded to the ScholarOne submission system.

We acknowledge that the paper exceeds the 4,000-word limit (current word count is 4,721); however, this was necessary to respond to the reviewer’s comments and to strengthen the paper.

We appreciate Dr. Ogrin’s review of the paper and the opportunity to make these revisions.

Sincerely,

Marie-Lee Yous, RN, MSc, PhD

Faculty of Health Sciences, School of Nursing
McMaster University
1280 Main Street West
Hamilton, Ontario, Canada
L8S 4K1
Email: yousm@mcmaster.ca
Phone: 905-920-4614